# Optimized RNA-Silencing Strategies for *Rice Ragged Stunt Virus* Resistance in Rice

**DOI:** 10.3390/plants10102008

**Published:** 2021-09-24

**Authors:** Severine Lacombe, Martine Bangratz, Hoang Anh Ta, Thanh Duc Nguyen, Pascal Gantet, Christophe Brugidou

**Affiliations:** 1PHIM Plant Health Institute, University Montpellier, IRD, CIRAD, INRAE, Institut Agro, 34090 Montpellier, France; martine.bangratz@ird.fr (M.B.); christophe.brugidou@ird.fr (C.B.); 2Plant Protection Research Institute (PPRI), Bac Tu Liem District, Hanoi 10000, Vietnam; anh742002@gmail.com; 3Agricultural Genetics Institute, Bac Tu Liem District, Hanoi 10000, Vietnam; nguyenthanhduc0212@yahoo.com.vn; 4UMR DIADE, Université de Montpellier, IRD, 34090 Montpellier, France; pascal.gantet@univ-montp2.fr

**Keywords:** RNA-silencing, virus resistance, RRSV, rice

## Abstract

*Rice ragged stunt virus* (RRSV) is one of the most damaging viruses of the rice culture area in south and far-eastern Asia. To date, no genetic resistance has been identified and only expensive and non-environmentally friendly chemical treatments are deployed to fight this important disease. Non-chemical approaches based on RNA-silencing have been developed as resistance strategies against viruses. Here, we optimized classical miRNA and siRNA-based strategies to obtain efficient and durable resistance to RRSV. miRNA-based strategies are involved in producing artificial miRNA (amiR) targeting viral genomes in plants. Classically, only one amiR is produced from a single construct. We demonstrated for the first time that two amiRs targeting conserved regions of RRSV genomes could be transgenically produced in *Nicotiana benthamiana* and in rice for a single precursor. Transgenic rice plants producing either one or two amiR were produced. Despite efficient amiR accumulations, miRNA-based strategies with single or double amiRs failed to achieve efficient RRSV resistance in transformed rice plants. This suggests that this strategy may not be adapted to RRSV, which could rapidly evolve to counteract them. Another RNA-silencing-based method for viral resistance concerns producing several viral siRNAs targeting a viral fragment. These viral siRNAs are produced from an inverted repeat construct carrying the targeted viral fragment. Here, we optimized the inverted repeat construct using a chimeric fragment carrying conserved sequences of three different RRSV genes instead of one. Of the three selected homozygous transgenic plants, one failed to accumulate the expected siRNA. The two other ones accumulated siRNAs from either one or three fragments. A strong reduction of RRSV symptoms was observed only in transgenic plants expressing siRNAs. We consequently demonstrated, for the first time, an efficient and environmentally friendly resistance to RRSV in rice based on the siRNA-mediated strategy.

## 1. Introduction

Rice is a staple food of over half the world’s population. Over 90% of the world’s rice is produced and consumed in the Asia-Pacific region. Due to population growth and urbanization, rice consumption will increase in the next years. Consequently, the intensification of production may increase. In parallel, threats to production will also increase with the emergence of diseases such as viral diseases. To effectively fight against these pathogens, it is necessary to be able to identify and deploy genes for effective resistance if they exist. One of the other ways is to use RNA-silencing strategies to inactivate the expression of targeted virus genes in the plant.

*Rice ragged stunt virus* (RRSV) belongs to the genus *Oryzavirus* within the family *Reoviridae*. It infects plants in the *Graminae* family, including rice *Oryza sativa*. It was first discovered in Indonesia and the Philippines in the 1970s [1]. The disease became prevalent in most rice growing countries in south and far-eastern Asia [2]. In Vietnam, this virus represents a major threat to rice production, as it is present in the Mekong Delta, which is the main rice producer representing 52% of the total Vietnamese production [3]. The main RRSV symptoms include stunting, twisted, and ragged leaves, which cause great losses to rice production [1]. RRSV is transmitted in a persistent manner by brown planthopper *Nilaparvata lugens* (BPH) after proliferation in the insect vector [4].

The RRSV genome consists of 10 double-stranded RNA (dsRNA) fragments with sizes ranging from 1.2 to 3.9 kb [5]. The segments are denoted as S1 to S10. The complete nucleotide sequences of all of them were determinate. In GenBank, partial and complete sequences of RRSV segments can be found from isolates collected in China, the Philippines, Vietnam, and India. Except for segment 4, which encodes two proteins, namely S4gp1 and S4gp2, all the other segments encode one protein named Sxgp1, where x indicates the fragment number. S8gp1 encodes an autocatalytic protease precursor that is self-cleaved into two proteins [6]. A total of 12 proteins are encoded by the RRSV genomes. The functions of some of them have been described. S4gp1 corresponds to RNA-dependent RNA polymerase (RdRP) [7]. The S6 fragment encodes a protein that has been described as both a movement protein [8] and an RNA-silencing suppressor [9,10]. S9gp1 has been characterized as a spike protein involved in vector transmission [11,12]. S7gp1 and S10gp1 have been described as non-structural proteins, whereas S5gp1 has been characterized as a minor structural protein [13,14]. No functional characterizations have been described for the other proteins. The dsRNA genome is encapsidated in icosahedral particles of approximately 35–70 nm in diameter [15].

To date, no genetic resistance has been discovered against RRSV in the natural diversity of rice and only chemical treatment against insect vectors is efficient in fighting against RRSV-associated disease. Due to its dramatic impact on rice production in the main rice-growing areas in south and far-eastern Asia, efficient and environmentally friendly solutions are crucially required to combat this disease. RNA-silencing is a natural eukaryotic process involved in RNA regulation and in efficient antiviral defense through the action of small RNA molecules [16]. There are two main classes of small RNAs: microRNA (miRNA) and small interfering RNA (siRNA). miRNAs are mainly 21-nucleotide (nt) RNA molecules produced from an endogenous RNA precursor displaying a double strand hairpin structure containing miRNA and its inverse complementary sequence miRNA*. miRNAs act as guides for the RNA-induced silencing complex (RISC) to target RNA degradation in a sequence-specific manner with some mismatch tolerance. SiRNAs are produced from a double-stranded RNA molecule synthesized by an RdRP and are 21-nt and 24-nt in size. As miRNAs, they act as guides for RISC to induce the degradation of RNA targets with perfect match sequence complementarity [17]. In plants, RNA-silencing mechanisms have been characterized as key processes in the regulation of plant development, in the maintenance of genome integrity, and in the defense against foreign nucleic acids such as viruses and transgenes [16]. This last property has been exploited to create virus-resistant transgenic plants. First, transgenic plants have been transformed with a sequence of the targeted virus, mainly from a gene encoding the coat protein [18,19,20,21]. Viral transgenic sequences are recognized by the plant defense mechanism as foreign nucleic acids that induce the RNA-silencing defense process through double strand RNA synthesis by RdRP. Consequently, viral siRNAs are produced from this transgene. Once a viral infection occurs, viral siRNAs already present in transgenic plants act to trigger virus degradation, inducing virus resistance. The most efficient way to produce viral siRNA from a transgene is to use inverted repeat DNA constructs producing hairpin RNAs that are recognized by the RNA-silencing defense machinery without the need for RdRP action. This process has been largely exploited for a large number of plant/virus pathosystems including rice since the early 2000s [22,23,24,25,26]. Moreover, chimeric hairpin constructs carrying sequences from different viruses have been used with success to induce multiple resistance against targeted viruses in transgenic plants [27,28,29]. It has been shown that the efficiency of hairpin RNA is dependent on the region targeted in the viral genome [30]. To improve the efficiency of siRNA-mediated resistance, multitargeting strategies have been developed. It has been shown that double transient transformation of *N. benthamiana* leaves with two siRNA constructs was more efficient in triggering *Olive mild mosaic virus* resistance than single transformation [31]. Similar to chimeric hairpin constructs acting on different viruses, chimeric constructs carrying sequences of different genes from a single virus have been used to maximize resistance efficiency. It has been shown that this chimeric hairpin construct improves virus resistance compared to hairpin constructs carrying sequences from a single gene [32].

The miRNA pathway has also been exploited to produce virus-resistant transgenic plants. This miRNA-based strategy concerns replacing the endogenous miRNA sequence in its precursor with an artificial miRNA (amiR) that targets the virus genome [33]. AmiR sequences are chosen in the virus genome according to the consensus of rules of base pairing for a functional amiR target interaction [34]. These rules include the tolerance of some mismatches, except at positions 2–11. Once introduced in plants, this amiR precursor is recognized by the endogenous miRNA machinery and produces the expected amiR that will target the viral genome in the case of infection [33]. Several transgenic plants expressing virus resistance based on the amiR strategy have been created [35,36,37,38]. However, it has been shown that viruses are able to counteract this amiR-mediated resistance. Indeed, they rapidly evolve to select mutations on the targeted sequences of their genome, impairing amiR recognition [39,40]. The use of two amiRs instead of one from two independent precursor constructs targeting a highly conserved viral genomic region has been shown to reduce this viral evolution strategy and improve the efficiency of amiR-mediated virus resistance [41].

Here, we aimed to develop original and efficient RNA-silencing-mediated resistance to RRSV in rice. For this purpose, we optimized and compared both siRNA and amiR strategies. For siRNA-mediated resistance, we used a chimeric double-strand construct carrying conserved sequences of three different genes of the RRSV genome. For the amiR approach, we aimed to compare the efficiency of a single amiR vs. double-amiR strategy. Most of the miRNA precursors described so far produce a single miRNA. However, a precursor producing two distinct miRNAs, specifically the osa-MIR159 precursor, was first described in rice and its conservation was validated in other species [42,43]. This precursor was used for the first time in this study to simultaneously produce two distinct amiRs against RRSV in rice transgenic plants. In parallel, transgenic plants expressing either one or the other amiR were also generated. All transgenic plants expressing siRNA, either single or double amiR, were challenged for RRSV resistance through BPH-mediated inoculation in South Vietnam. This work allowed for the generation of efficient and environmentally friendly RRSV resistance in rice for the first time.

## 2. Results

### 2.1. amiR Selection

Among the 11 ORFs encoded by the RRSV genome, the functions have been more or less described for eight of them (Figure 1). To select an amiR target, we expected that the most efficient resistance would be obtained for a targeted gene encoding a protein with a key function in virus development. RRSVs6gp1 was selected due to its dual key function as a movement protein and silencing suppressor [8,9,10]. Moreover, since our strategies involve RNA-silencing, targeting this segment would avoid the suppression of the strategies we wanted to apply (Figure 1).

First, the microRNA designer tool was used to identify potential amiRs considering the following parameters: a unique target in the RRSV genome and zero off-target in the rice genome. The two first best candidates (amiR1 and amiR2) were retained. Our goal was not only a strong but also a durable and large-spectrum RRSV resistance. As previously demonstrated [41], we assumed that highly conserved amiR targets in the RRSVS6gp1 sequence diversity would be an accurate candidate to reach this goal. Consequently, to validate the artificial miR1 and miR2 (amiR1 and amiR2) proposed by the microRNA designer tool, we evaluated the conservation of selected amiR targets among the RRSV isolate genetic diversity. Nucleotide sequence alignments were performed between (1) the inverted complementary sequences of either amiR1 and amiR2 (amiR1 comp and amiR2 comp) and (2) and the RRSVs6gp1 sequences available in the GenBank resource (Figure 2). Seven RRSVs6gp1 sequences from different origins in Southeast Asia aligned with the selected amiR comp sequences. For amiR1, one mismatch was observed at position 1 on the 5’side of amiR1 for all seven sequences. A bulged nt at position 14 was also observed for six out of the seven sequences (Figure 2A). In the case of amiR2, only one mismatch at position 1 on the 5’side of amiR2 was detected for all seven sequences (Figure 2B). These polymorphisms fit the consensus rules of base pairing established for a functional plant miRNA-target interaction: little tolerance of mismatches at positions 2 to 13 and no tolerance of mismatches at positions 9 to 11 [34]. Cases of bulged nucleotides involving one of more nucleotides were also reported not to have impaired the functionality of these miRNAs in plants [34,44]. The selection of amiR1 and amiR2 was validated to generate amiR constructs.

### 2.2. Validation of the Functionality of the Rice miRNA Precursor to Produce Double Artificial miRNA

Classically, transgenic rice expressing artificial miRNA was produced using a rice endogenous miRNA precursor producing osa-MIR528. amiRs replaced endogenous osa-MIR528 and osa-MIR528* [45,46,47]. This strategy was used to produce two precursor constructs expressing either RRSV amiR1 or amiR2 (pC:amiR1 and pC:amiR2). In parallel, another endogenous rice miRNA precursor, namely the osa-MIR159 precursor, that has been characterized as able to simultaneously produce two distinct miRNAs in rice was used [42]. amiR1 and amiR2 were integrated in this double miRNA precursor to replace both endogenous miRNA/miRNA* couples to generate the pC:amiR1/2 construct. As this double precursor has never been used to produce artificial miRNA, we wanted to validate its function in planta before starting a long-lasting stable rice transformation procedure. This pC:amiR1/2 was transiently expressed in *Nicotiana benthamiana* leaves to verify whether it could effectively be recognized by the plant miRNA machinery to produce the two expected artificial miRNAs. Rdr6 invert repeat transgenic *N. benthamiana* that is unable to induce siRNA defense machinery against foreign nucleic acids was used to exclude the possibility that the small RNA detected could be produced by this defense machinery and not by the miRNA machinery. pC:amiR1 and pc:amiR2 were used as positive controls (Figure 3). As anticipated, both the control pc:amiR1 and pc:amiR2 constructs produced their expected miRNAs as shown by the specific signals. Signals were also detected in the pC:amiR1/2 sample with both amiR1 and amiR2 complementary probes (Figure 3). These results demonstrate that our double artificial miRNA construct can be recognized by the miRNA machinery in planta to produce the expected amiR.

### 2.3. Selection of RRSV Sequences for siRNA Synthesis

A hairpin construct generated to produce viral siRNAs was designed to target three different RRSV genes. This multitargeting is expected to increase the efficiency of viral resistance, as previously demonstrated [31,32]. We based the choice of two of the target genes on their fundamental function in viral development. As for artificial miRNA selection, RRSVs6gp1 encoding movement proteins and silencing suppressors were chosen [8,9,10]. RRSVs9gp1 coding for spike proteins was been chosen because it plays a role in virus transmission [11,12]. Finally, the third gene S10gp1 was randomly chosen among genes without precise function.

As for amiR selection, the conservation criteria were taken into account to achieve a broad spectrum and durable resistance. To select three regions for siRNA production among RRSVs6gp1, RRSVs9gp1, and RRSVs10gp1 sequences, BLAST searches were performed with each of the three genes. These approaches revealed relatively high conservations at the nucleotide level for RRSV-collected sequences (Figure 4). For RRSVs6gp1 and RRSVs9gp1, nucleotide identities for the seven identified sequences were between 99% and 100%, and between 94% and 100%, respectively. For RRSVs10gp1, the six identified sequences displayed nucleotide identities between 99% and 100%. For each targeted gene, 150-nt sequences were randomly chosen (Figure 1). The RRSVs6gp1 (Figure 4A) and RRSVs10gp1 (Figure 4C) selected sequences showed extremely reduced diversity between sequences from different RRSV isolates from southeast and south Asia. The selected RRSVs9gp1 sequence displayed a stronger diversity but only with two out of the seven selected sequences from different origins in southeast and south Asia (Figure 4B). These two diverse sequences (HM125567 and L38900) were from China and India, with their diversity being quite moderate (only 11 distinct nucleotides out of 150 compared to the conserved five other sequences). These three selected sequences were used to create inverted repeat constructs in the pANDA vector (pANDA:siRRSV).

### 2.4. Accumulation of Expected amiR and siRNA in Transgenic Plants

Each amiR and siRNA construct was used to transform the Nipponbare rice variety. An empty pCambia pC5300UBI/NOS was used for as a negative control. For each transformation, two homozygous T2 transgenic lines were selected: pCamiR1_2.1_, pCamiR1_20.1_, pCamiR2_2.2_, pCamiR2_22.4_, pCamiR1/2_9.1_, and pCamR1/2_12.3_, as well as pANDA:siRRSV_39.9_, pANDA:siRRSV_40.3_, and pANDA:siRRSV_41.9_. To verify the expected amiR and siRNA accumulation in the selected transgenic lines, northern blots of leaf RNA were hybridized with the corresponding probes. Plants transformed with pCambia empty vectors (pC:empty) and Nipponbare lines were used as negative controls for the absence of amiR and siRNA accumulation, respectively (Figure 5). As expected, neither amiR1 nor amiR2 accumulated in the pC:empty negative control. None of the amiR1 lines, namely pC:amiR1_2.2_ and pC:amiR1_20.1_, accumulated amiR1, whereas amiR2 accumulation was detected in both amiR2 lines, namely pC:amiR2_2.2_ and pC:amir2_22.4_ (Figure 5A). In the two selected lines transformed with the construct aiming to produce both amiR1 and amiR2, accumulation of these two amiRs was detected (Figure 5A). These results suggest that the amiR1 construct would be inactive. Only the amiR2 and amiR1/2 constructs were functional to produce amiR2 and both amiR1 and amiR2, respectively. The inactivity of pC:amiR1 may not have been due to the Osa-MIR528 precursor backbone, as the same backbone was used for the amiR2 construct. It is unlikely that amiR1 inefficiency would be due to amiR1 instability, as amiR1 is able to accumulate when it is produced from the double amiR1/2 precursor. The amiR1 construct could have been partially destroyed during the rice transformation.

siRNA accumulation corresponding to each of the three segments of the multiple siRNA construct was evaluated by independent hybridizations with three probes, namely the S6, S9, and S10 probes, covering the three segments (Figure 5B). As expected, no signal was detected in the negative control corresponding to the Nipponbare non-transformed line. Surprisingly, no signal was detected for the pANDA:siRRSV_39.9_ line with any of the three probes, suggesting that the siRRSV construct is inactive in this line. Interestingly, for the pANDA:siRRSV40.3 line, the signal was detected only with the S6 probe, suggesting that double-stranded RNA can only be formed on the S6 segment. The pANDA:siRRSV41.9 line accumulated siRNAs from the three segments, as signals were detected with the three probes (Figure 5B). As proposed above for the amiR1 construct, the partial or total inefficiency of the multiple siRRSV constructs may have been due to partial degradation of the transgenic construct during the rice transformation for the pANDA:siRRSV_39.9_ and pANDA:siRRSV_40.3_ lines.

### 2.5. Evaluation of RRSV Resistance

Taichung Native-1 (TN1) is a variety classically used as a pathogen-susceptible reference variety [48]. It belongs to the *Oryza sativa indica* sub-species, which is not adapted for genome transformation. Here, transformations were performed in Nipponbare (Nip), which belongs to the *Oryza sativa japonica* sub-species. This is one of the most commonly used varieties for rice transformation due to its good transformation efficiency. To evaluate the transgenic plants produced here for RRSV resistance, we first had to evaluate Nip behavior in response to RRSV infection compared to TN1. Height reduction is one of the RRSV symptoms that is quite easy to measure. First, TN1 and Nip heights were compared without any virus infection constraint under insect-free conditions during 27 days after infection (DAI) (Figure 6). The results showed that both varieties acted similarly for the height parameter under the non-virus infection constraint. Under RRSV infection, the expected size reduction was observed for the TN1 reference variety, with a size reduction of more than half at 27 DAI compared to the non-infected condition (approximately 30 cm vs. 70 cm). The same magnitude of size reduction was also observed for infected Nip (Figure 6). These observations reveal that Nip is susceptible to RRSV and indicate that this susceptibility leads to an important size reduction of more than half compared to the non-infected condition four weeks after infection. This reduced size phenotype is quite easy to follow to identify susceptible plants. Visual observations were used for the following experiments to evaluate susceptible/resistant phenotypes until eight weeks after inoculation. Plants with a reduced size of around 50% at 27 dpi were considered as RRSV-sensitive.

Homozygous transgenic lines were evaluated for their resistance to RRSV transmitted by BPH. Several genotypes were used as positive susceptible controls: TN1 as the susceptible reference variety and Nip as the variety chosen for both transformation procedures and for lines transformed by the empty vector (pC:empty). For each line, approximately 100 plants were transplanted, except for pC:amiR2/2.2 plants, with only 16 transplanted plants (Table 1). This reduced number of plants is due to the low seed germination for this line. This could be due to the transgene presence. Consequently, this line would not be an accurate candidate because of this non-agronomical behavior. The RRSV symptom observed was a height reduction 40 days after RRSV transmission by BPH (Figure 6, Table 1). Transplanted plants displayed similar and normal development phenotypes and only a few plants died before RRSV infection (Table 1). These observations suggest that transgene insertion has no major effect on plant development, except for pC:amiR2/2.2 transgenic plants.

For the TN1 susceptible reference variety, the proportion of plants displaying symptoms was 44.95%. This proportion agrees with the expected proportion based on RRSV transmission by its vectors under controlled conditions in the PPRI station, south Vietnam. For the wild-type Nip control, the proportion of symptomatic plants was reduced by 36.89%. As the viruliferous BPH population used for the RRSV inoculations was the same for all the lines tested, the decreased proportion of symptomatic plants observed between Nip and TN1 might have been due to a lesser attraction of BPH for the Nip genotype. As expected, control plants transformed with the pC:empty vector displayed similar proportions of symptomatic plants (39.39%). Lines transformed with the pC:amiR1 construct (38% and 40.63%) did not show any decrease in the proportion of symptomatic plants compared to the Nip or the pC:empty controls. This observation is in agreement with the demonstrated absence of amiR1 accumulation (Figure 5A). For one of the pC:amiR2 lines (pc:amiR2_2.2_), this proportion dropped to 6.25%. However, only 16 plants were evaluated, which we considered as insufficient for a solid interpretation. Moreover, despite this low percentage, this line displayed a very low germination rate, making it not a suitable candidate for agronomical uses. For the other pC:amiR2 line (pc:amiR2_22.4_), the proportion of symptomatic plants was slightly reduced to 20% compared to the Nip or pC:empty controls. These observations suggest that amiR2 has a slight effect on the RRSV symptom development. Lines transformed by pC:amiR1/2 that produced both amiR1 and amiR2 displayed 33% and 33.02% of the symptomatic plants. These proportions, which were similar to the control plants, suggest that the production of amiR1 and amiR2 together has no effect on the reduction of the RRSV symptom development. Moreover, the slight effect of amiR2 as observed on the pC:amiR2_2.2_ transgenic plants was not reported for the lines transformed by pC:amiR1/2. This suggests that this slight effect of amiR2 on the RRSV symptom development would not be robust.

Concerning plants transformed with the RRSV siRNA construct (pANDA:siRRSV), two out of the three independent transgenic lines tested displayed strong reductions in the proportion of symptomatic plants, with 6.67% and 2.02% for pANDA:siRRSV_40.3_ and pANDA:siRRSV_41.9_ plants, respectively (Table 1). For the other pANDA:siRRSV (pANDA:siRRSV_39.9_), this proportion was similar to that of the control Nip plants (34.62% vs. 36.89%). This observation agrees with the fact that any expected siRNA accumulated in the pANDA:siRRSV_39.9_ line (Figure 5B). Therefore, the strong reduction in symptom development is correlated with the accumulation of siRNA. The difference in the proportions of symptomatic plants between pANDA:siRRSV_40.3_, accumulating siRNA only from the S6 fragment, and pANDA:siRRSV_41.9_, accumulating siRNA from all three fragments, was not significant. Consequently, siRNA produced from one gene fragment is sufficient to obtain significant resistance to RRSV.

Taken together, these results show strong RRSV resistance only for transgenic plants expressing viral siRNA, whereas amiR does not seem to have a major effect.

## 3. Discussion

In this work, we compared the efficiency of original RNA-silencing strategies to induce resistance in rice against one of the most damaging viruses of south and far-eastern Asia.

We first optimized the amiR strategy, considering, for the first time, a rice miRNA precursor that produces two amiRs instead of one. Indeed, it has been previously shown that the use of two amiRs instead of one improves the efficiency of amiR-mediated virus resistance by reducing the virus evolution probability [41]. Polycistronic amiR constructs able to produce several amiRs that target viruses in wheat and barley have been used [22,49]. One of these constructs is based on the rice Osa-MIR395 precursor, which displays five hairpin structures. Each of them produces one amiR targeting *wheat streak mosaic virus* in wheat [22]. The other polycistronic construct is based on the barley hvu-MIR171 precursor, producing a single miRNA. A tandem of three repeats of this precursor was built to produce three amiRs targeting the *Wheat dwarf virus* once introduced into the barley genome [49]. Here, we used the osa-MIR159 precursor, which is an endogenous rice miRNA precursor that naturally produces two miRNAs from a single hairpin structure [42]. We validated its ability to produce the two expected amiRs, namely amiR1 and amiR2, transiently in *N. benthamiana* and in stably transformed rice plants. Unlike the polycistronic amiR precursors described above, amiR1 and amiR2 were produced from the same hairpin structure. This suggests that in our case, a single hairpin structure needs to be formed, whereas in the works presented by Fahim et al. [22] and Kis and al. [49], several hairpins had to be created for the polycistronic amiR constructs to be effective. This observation suggests that the structure required to produce the expected multiple amiRs would be the easiest to obtain for the osa-MIR159-based amiR precursor than for the other polycistronic amiR precursors. Consequently, we could speculate that the osa-MIR159 precursor would be more efficient for a large-scale production of transgenic rice producing double amiR.

Despite amiR selection in conserved RRSV regions and effective amiR accumulation, transgenic rice producing amiR2 or both amiR1 and amiR2 did not show any efficient resistance to RRSV. This suggests that the virus genome was able to evolve to counteract this amiR-mediated resistance. Sequencing of the amiR target region in the RRSV population multiplied on amiR transgenic rice could confirm this hypothesis. Even if multiple and simple amiR strategies demonstrated to be efficient in triggering virus resistance in different plant/virus pathosystems including rice [22,35,36,37,38,49,50], they may not be adapted for RRSV resistance. However, it has been shown that in the case of siRNA-mediated strategies, its efficiency is dependent on targeted genes [30]. We cannot exclude the possibility that RRSVs6gp1 targeted by amiR1 and amiR2 would not be sufficient to induce resistance.

Even if our multiple amiR strategy failed to trigger RRSV resistance, it would be efficient for simply and efficiently producing two amiRs. In a more general context than virus resistance, using multiple amiR strategies could present great interest regarding the targeting of multiple non-family genes. Moreover, obtaining multiple amiRs from a single construct represents a simple and cost-effective strategy. The Osa-MIR159-based precursor could be used for this purpose.

The siRNA-mediated strategy we developed here was optimized to target conserved regions of three different RRSV genes. The construct we used carries these three regions in tandem and in invert repeat orientation. The efficiency of siRNA production from the three segments was validated in transgenic rice in only one line out of the three selected lines. The other fragments produce either siRNAs from only one fragment or no siRNAs at all. Finally, we demonstrated that transgenic lines producing siRNAs from one or from the three fragments display a strong improvement in their resistance against RRSV. Individual plant measuring or RRSV-specific RT-PCR could be performed in the future to confirm pANDA:siRRSV_40.3_ and pANDA:siRRSV_41.9_ RRSV resistance compared to Nipponbare. Although we observed a slight increase in resistance with the three fragments, the difference in the proportion of symptomatic plants between the two resistant lines was not significant. We speculate that RRSV resistance would be more efficient in the pANDA:siRRSV_41.9_ line, accumulating siRNAs from all three fragments, than in the pANDA:siRRSV_40.3_ line, accumulating siRNAs only from the S6 fragment. The number of tested plants should be increased to evaluate this improved RRSV resistance hypothesis.

Although it is difficult to interpret the additive effect of siRNA expression from three gene fragments on the improvement of resistance, RRSV durability could be tested, as we expect RRSV resistance to be more durable in the pANDA:siRRSV_40.3_ lines than in the pANDA:siRRSV_41.9_ lines.

Multiplexing of viral segments has been shown to be efficient in targeting several viruses. This strategy was used with success in *N. benthamiana* against four tospoviruses, namely *tomato spotted wilt virus* (TSWV), *groundnut ringspot virus* (GRSV), *tomato chlorotic spot virus* (TCSV), and *watermelon silver mottle virus* (WSMoV) [27]. More recently, siRNA-mediated multiple virus resistance has also been used in rice against two viruses causing rice tungro disease [28]. In 2017, Kumar and colleagues compared the efficiency of *Mungbean yellow mosaic India virus* (MYMIV) resistance induced by siRNA-based constructs targeting one or two viral genes in cowpea. Transgenic cowpea carrying the double siRNA construct displayed an improved resistance compared to the wild type and lines transformed with a construct targeting a single gene [32]. Here, by targeting three RRSV genes, we optimized the probability of obtaining efficient RRSV resistance in rice.

A multiple siRNA-mediated strategy has also been used with success to simultaneously knockdown six non-family genes in *Arabidopsis thaliana* [51]. The techniques reported here to simultaneously target three RRSV genes from a single transgene could be exploited to silence multiple viruses or endogenous genes in rice in a simple and cost-effective manner.

In conclusion, the work reported here introduces important perspectives for the release of transgenic rice displaying strong resistance to one of the most damaging viruses in the main rice-growing countries in south and far-eastern Asia. The use of this transgenic rice would drastically reduce the use of expensive and non-environmentally friendly chemical treatments to fight against this disease. As reviewed recently in Cisneros and Carbonell [52], strategies based on miRNAs, trans-acting siRNAs, and siRNAs are strongly expected to simultaneously produce several artificial small RNAs. The double miR and multiple siRNA techniques covered here can be considered as some of these improved strategies and could be exploited in a more general context, as opposed to virus resistance, to simply and efficiently silence multiple target genes in rice.

## 4. Material and Methods

### 4.1. Generation of RRSV amiR Precursor Constructs

RRSV-specific amiRs were selected from the RRSVs6gp1 sequence AF020337 using Web MicroRNA Designer 3 (http://wmd3.weigelworld.org/cgi-bin/webapp.cgi, accessed on 24 September 2021). The specific parameters were one unique target in the RRSV genome and zero off-targets in the rice genome. After the diversity analyses presented in the Results section, two sequences, namely amiR1 (5′TAGAGACTGCGTTACGATCAA3′) and amiR2 (5′TCGGTGTACCATTCCAGCCTT3′), were selected. amiR constructs producing a single amiR were generated using the *Oryza sativa* Osa-MIR528 precursor as backbone, wherein endogenous miR528/miR528* was replaced by either amiR1/amiR1* or amiR2/amiR2* [47]. amiR1* and amiR2* correspond to the complementary sequence of the corresponding amiR with 3 mismatches noted in bold (amiR1*: ^5′^TTGATGGTATCGCACTCTCTA^3′^ and amiR2*: ^5′^AAGGCAGGATTGGTTCACCGA^3′^). amiR construction producing both amiRs was done using a fragment corresponding to the *Oryza sativa* Osa-MIR159 precursor as the backbone. This precursor produces two miRs from a single hairpin [42]. miR159a/miR159a* and miR159b/miR159b* from the endogenous sequence were replaced by amiR1/amiR1* and amiR2/amiR2*. Sequences of all amiR precursors were de novo synthetized in a pUC18 vector by Genecust Company with the *Bam*HI and *Ass*65I restriction sites at their 5′ and 3′ extremities, respectively (Luxembourg). amiR precursors were then cloned between the UBI promoter and NOS terminator sequences in the pC5300UBI/NOS vector from the rice functional genomics platform (CIRAD, France) at the *Bam*HI and *Acc*65I restriction sites. Four pC5300 constructs were obtained: pC:amiR1 and pCamiR2 with precursors carrying either amiR1 or amiR2 sequences; pC:amir1/2 with precursors carrying both amiR1 and amiR2 sequences; and pC:empty control without any inserted fragments. All these constructs were transformed into *Agrobacterium* strain EHA105 by electroporation.

### 4.2. Vector Construction for RRSV siRNA Production

Sequences from RRSVs6gp1 (AF020337), RRSVs9gp1 (GQ329711), and RRSVs10gp1 (U66712) were used to choose conserved 150-nt fragments. First, a BLAST search was performed to evaluate the nucleotide diversity among related sequences. Due to high conservation, 150-nt fragments were randomly chosen in each of the three genes. These fragments (450-nt tandem) were de novo synthetized in a pUC18 vector by Genecust Company (Boynes, France) with att recombination sites for further gateway cloning (Luxembourg). Then, the tandem sequence was first inserted in the gateway cassette of a pDONR207 vector at the BP recombinase site (Invitrogen, Carlsbad, CA, USA). Then, using LR recombinase (Invitrogen), the tandem sequence was transferred into the two gateway cassettes of the pANDA vector in sense and antisense orientations [53]. The resulting construct, specifically pANDA:siRRSV, was transformed into *Agrobacterium* strain EHA105 by electroporation.

### 4.3. Transient Expression Assay of amiR Precursors in Nicotiana Benthamiana

*Agrobacterium* solutions carrying pC:empty, pC:amiR1, pC:amiR2, and pC:amiR1/2 were prepared as described in [54]. Concentrations of bacterial solutions were homogenized at OD600 = 0.5. For the transient expression assay, RDR6i *N. benthamiana* that displayed silencing against its RDR6 gene was chosen to avoid the RDR6-dependent defense-silencing pathway from acting on pCambia constructs [54]. Agroinfiltrations were performed on *N. benthamiana* leaves of 4-week-old plants.

### 4.4. Rice Transformation Procedure

The Nipponbare *japonica* rice variety was used for the transformation procedures due to its good transformation potential [55]. The pC:empty, pC:amiR1, pC:amiR2, pC:amiR1/2, and pANDA:siRRSV transformations into the Nipponbare background were performed as described in [55].

### 4.5. Selection of Homozygous Transgenic Rice Plants by PCR and Southern Blot

Bits of leaves were collected on T0-transformed plants 6 months after the beginning of the transformation process before their transfer to soil. Genomic DNA was extracted using the MATAB/CETAB method [56]. The presence of the transgene was detected by PCR on the kanamycin resistance gene of the pCambia and pANDA transgenic constructs using F_kana_ (^5′^ATGGCTAAAATGAGAATATCACC^3′^) and R_kana_ (^5′^ATGTCATACCACTTGTCCGCCC^3′^). Positives plants were transferred to soil. New leaf sampling of approximately 0.5 g was performed three weeks later for DNA extraction using the MATAB/CETAB method. Southern blotting was performed as described in Sallaud et al. (2003) from 2 µg of DNA digested with *Hind*III. After migration on 0.8% agarose and after the alkaline transfer of DNA to a Hybond N+ membrane (GE Healthcare, Chicago, IL, USA), membranes were hybridized with a kanamycin probe (F_kana_ R_kana_ PCR fragment) labeled with α-[^32^P]-dCTP using the Rediprime II DNA labeling system (GE Healthcare). Autoradiographies were obtained by exposure to phosphor screens, read by the Typhoon imaging system (GE Healthcare).

For each of the 3 miRNA constructs and their control (pC:amiR1, pC:amiR2, pC:amiR1/2, and pC:empty) as well as the siRNA construct (pANDA:siRRSV), between 7 and 10 T0 lines were selected for T1 seed harvesting. Mono insertion of the transgenes was verified by kanamycin-resistance screening (3:1 ratio, resistant:susceptible) on T2 seedlings. T2 seeds were obtained from antibiotic-resistant lines. Antibiotic selection was performed on these T2 seeds to identify homozygous lines for the transgene that should display 100% antibiotic resistance. Finally, for each amiR construct, two independent homozygous lines were selected (pCamiR1_2.1_, pCamiR1_20.1_, pCamiR2_2.2_, pCamiR2_22.4_, pCamiR1/2_9.1_, and pCamR1/2_12.3_). For its control (pC:empty), only one homozygous line was kept. For the siRNA construct, three independent homozygous lines were chosen (pANDA:siRRSV_39.9_, pANDA:siRRSV_40.3_, and pANDA:siRRSV_41.9_).

### 4.6. Small RNA Northern Blots

Total RNA was isolated from *N. benthamiana* and rice leaves using TRIzol Reagent (Ambion). Small RNA blots and northern hybridization were performed as described in [57]. Complementary oligonucleotides of amiR1 (^5′^TTGATCGTAACGCAGTCTCTA^3′^) and amiR2 (^5′^AAGGCTGGAATGGTACACCGA^3′^) were used as probes to detect amiR1 and amiR2 accumulation, respectively. The complementary oligonucleotide of osa-Mir159 (^5′^TGCAGCTCCTGGGGCATGCAA^3′^) was used to hybridize miR159 as an internal quantity control. These oligonucleotide probes were end-labelled with γ-[^32^P]-ATP using T4 polynucleotide kinase (Promega). To detect RRSV siRNA, probes corresponding to each of the three fragments were produced by PCR on the pANDA:siRRSV vector using F_S6_ (^5′^GTTCGTGCCCGAGTACGTTG^3′^) and R_S6_ (^5′^GATATCATTTCCATAGACGC^3′^) to amplify the RRSVs6gp1 fragment; F_S9_ (^5′^CTATTTGTCAATCGCGGCGA^3′^) and R_S9_ (^5′^TCCAGCGTCTAGTCCCGTAT^3′^) to amplify the RRSVs9gp1 fragment; and F_S10_ (^5′^GCCACAGGTAAGCGTACATT^3′^) and R_S10_ (^5′^AATCAGAATATAATCCATCA^3′^) to amplify the RRSVs10gp1 fragment. siRNA probes were labeled with α-[^32^P]-dCTP using the Rediprime II DNA labeling system (GE Healthcare). Small RNA hybridization was performed overnight at 40 °C in PerfectHyb Buffer (Sigma, Kawasaki, Japan), followed by a single wash at 50 °C in 2X SSC and 0.1% SDS for 10 min. Autoradiographies were obtained by exposure to phosphor screens, read by a Typhoon imaging system (GE Healthcare).

### 4.7. Evaluation for RRSV Resistance

The evaluation of RRSV resistance for the Taichung Native-1 (TN1) variety as a susceptible reference, for Niponbarre as a wild-type control, and for transgenic plants was performed at the Plant Protection Research Institute (PPRI) station in south Vietnam. Infected RRSV rice plants with typical symptoms were collected in the Tien Giang and Long An provinces from Mekong Delta (Vietnam). RRSV infection was checked by RT-PCR using specific primers. The absence of Rice grassy stunt virus (RGSV) was also verified by RT-PCR with specific primers as this virus, transmitted by BPH, is also present in Mekong Delta. The protocol used for RRSV infection is summarized in Figure 7. Healthy nymphs of the second to third instar of BPHs were brought onto RRSV-infected rice leaves at approximately 5 to 10 nymphs per plant depending on the size of the infected rice plant. Individual plants with BHP nymphs were put into individual plastic boxes with holes sealed with an insect-proof net. These nymphs were kept for at least 10 days for virus acquisition and propagation. At the same time, plants to be tested were sown after soaking their seeds for 48 h. Then, only adults were selected on infected plants for inoculation on 10-day-old plants. For this, one single BPH was added into one test tube containing one 10-day-old plant for 24 h. After 24 h, the BPH was kept in a test tube and the rice plant was replaced by another one to be tested. The inoculations were performed serially until the BPH died. After inoculation, the rice plants were transplanted into experimental plots with insect-proof cages under natural rice growing conditions at the PPRI station in south Vietnam. Plants inoculated from the same BPH were identified and transplanted in a separated row. Plant height was followed until 30 days after infection (DAI), corresponding to 40 days after germination (Figure 7).

## Figures and Tables

**Figure 1 plants-10-02008-f001:**
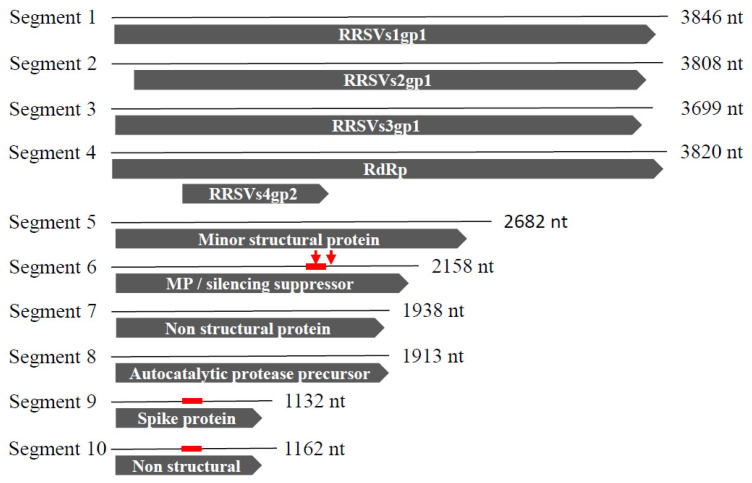
Rice ragged stunt virus (RRSV) multipartite genome. Each of the 10 RNA segments is represented. Black lines correspond to RNA genomes and gray arrows represent ORFs. Names or functions of ORFs, when they are known, are noted in white. The name and size of each RNA segment are indicated. The positions of artificial miRNA targets are indicated by red arrows on segment 6. Sequences selected to construct the chimeric RRSV double strand are indicated by red lines on segments 6, 9, and 10.

**Figure 2 plants-10-02008-f002:**
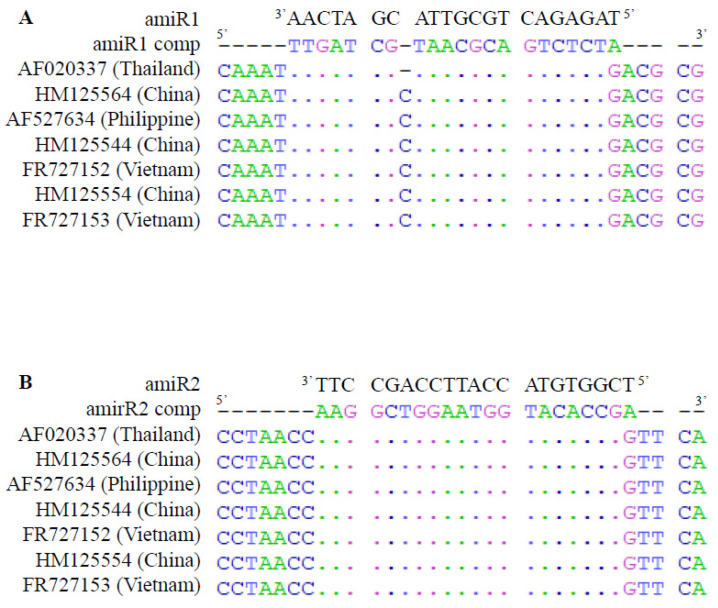
Alignment of amiR1 and amiR2 target sites in RRSV genomes from the GenBank resource. amiR1 (**A**) and amiR2 (**B**) complementary (comp) sequences were aligned to RRSVs6gp1 sequences deposited in GenBank. The accession number and origin of each sequence are indicated. Nucleotide identity and deletions are represented by a dot and a small line, respectively. The sequences of mir1 and mir2 are indicated in black in the 3′–5′ orientation.

**Figure 3 plants-10-02008-f003:**
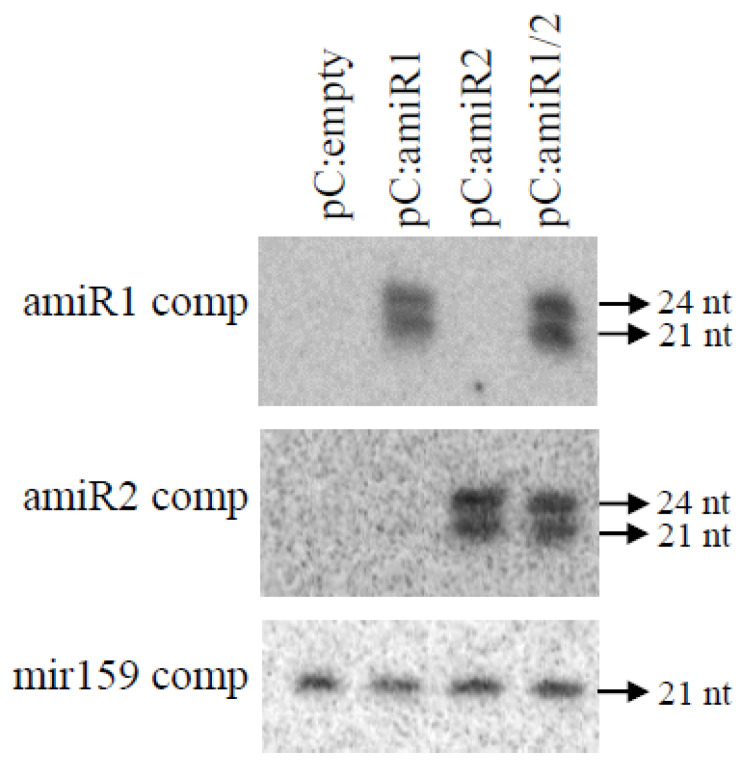
Northern blot detecting amiR accumulation in *N. benthamiana* leaves. *N. benthamiana* leaves were transiently transformed with empty vectors or pCambia vectors containing the amiR1 construct (pc:amiR1), amiR2 construct (pc:amiR2), or amiR1/2 construct (pC:amiR1/2). Northern blots were hybridized with amiR1 or amiR2 complementary sequences as probes (amiR1 comp and amiR2 comp). mir159 hybridization with a mir159 complementary probe (mir159 comp) was used as an equal loading control. The sizes of the detected bands are indicated.

**Figure 4 plants-10-02008-f004:**
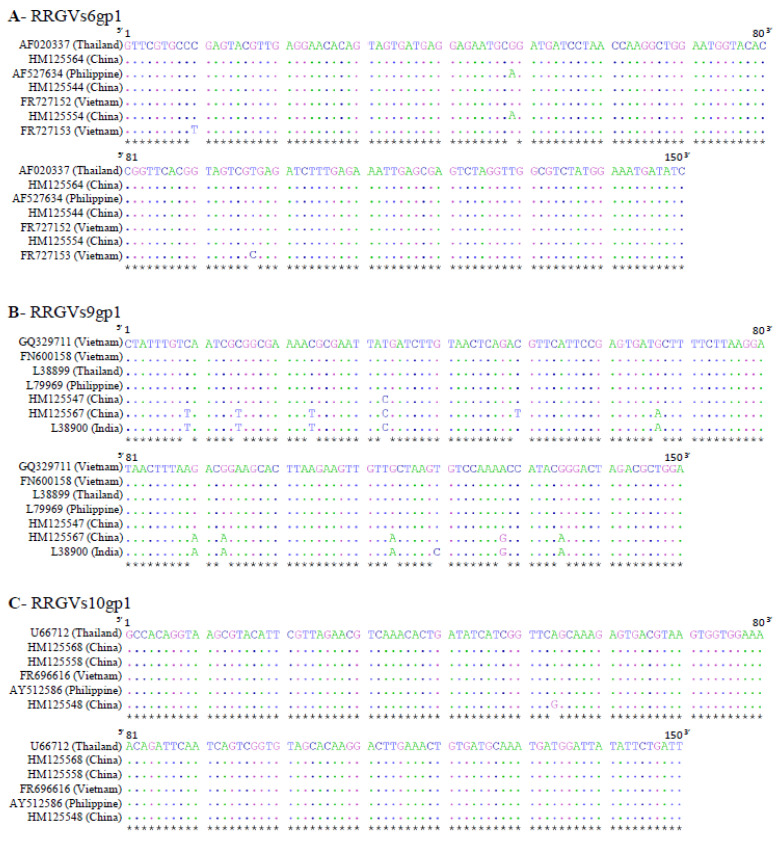
Alignment of selected siRNA regions from RRSVs6gp1 (**A**), RRSVs9gp1 (**B**), and RRSVs10gp1 (**C**) with RRSV sequences from the GenBank resource. The accession number and origin of each sequence are indicated. Nucleotide identity is represented by a dot. Stars at the bottom indicate the nucleotide identity among all the selected sequences. Numbers above sequences indicate nucleotide positions and orientations.

**Figure 5 plants-10-02008-f005:**
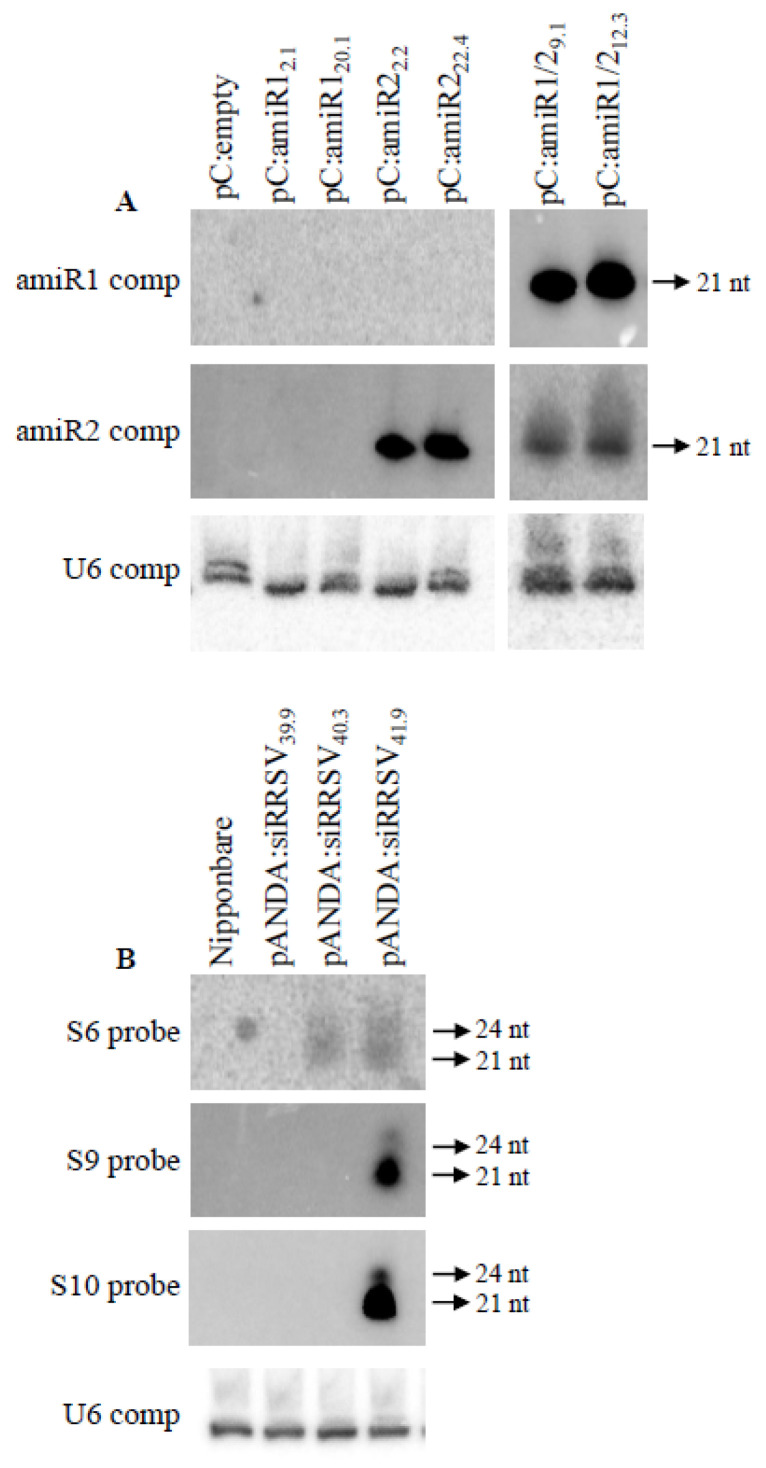
Northern blot detecting amiR (**A**) and siRNA (**B**) accumulation in selected homozygous transgenic lines. Total RNA was extracted from three-week-old rice leaves. (**A**) amiR accumulation was detected by hybridization with the amiR1 or amiR2 complementary sequences as probes (amiR1 comp and amiR2 comp). A transgenic line transformed with an empty construct was used as a negative control (pC:empty). (**B**) siRNA accumulation was detected by hybridization with PCR fragments corresponding to the RRSVs6gp1 fragment (S6 probe), the RRSVs9gp1 fragment (S9 probe), and the RRSVs10gp1 fragment (S10 probe). A non-transformed Nipponbare line was used as a negative control (Nipponbare). For both blots, a U6 hybridization with the U6 complementary probe (U6 comp) was used as an equal loading control. The sizes of the detected bands are indicated.

**Figure 6 plants-10-02008-f006:**
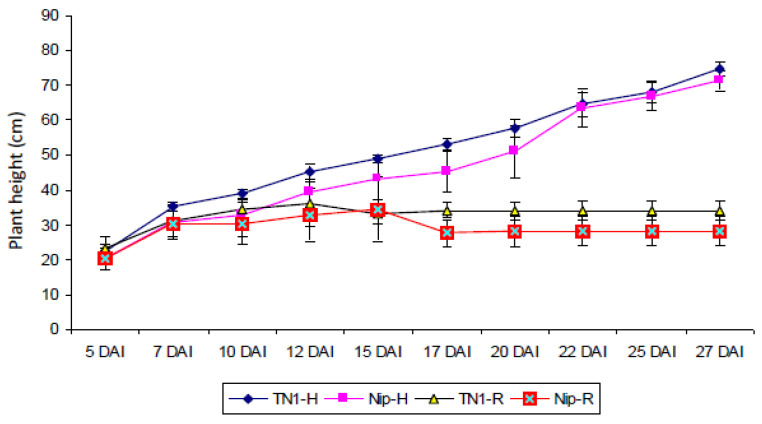
Comparison of plant size evolution between TN1 reference and Nip varieties in non-infected and RRSV-infected contexts. For each variety and condition, the size of approximatively 100 plants was followed for four weeks. RRSV inoculation was performed on 10-day-old plants. Non-infected healthy conditions are noted as –H (TN1-H and Nip-H). The infected condition is noted as –R (TN1-R and Nip-R). Time is reported in days after infection (DAI). The means of plant sizes are reported for each condition and error bars represent the standard deviation. Plant size is noted in cm.

**Figure 7 plants-10-02008-f007:**
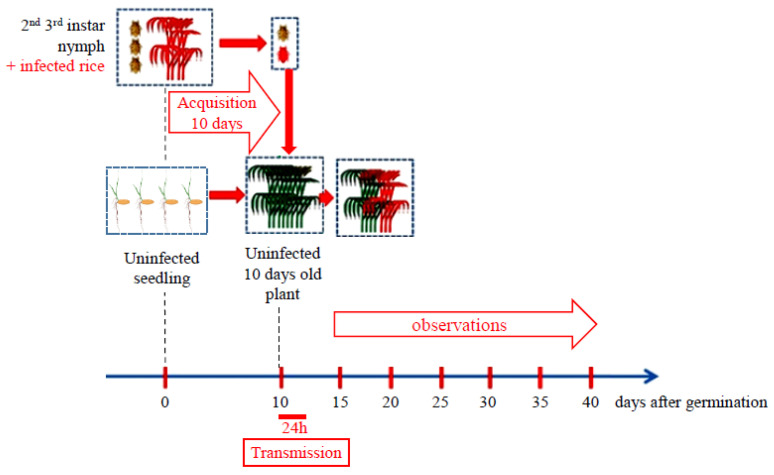
Illustration of the RRSV infection process developed in PPRI for RRSV resistance evaluation. This process consists of three steps, as noted in red. The first step is RRSV acquisition by BHPs over 10 days. The second step is RRSV transmission by viruliferous BHP to 10-day-old plants over 24 h. The last step is plant height observation over 30 days following infection.

**Table 1 plants-10-02008-t001:** Infectivity assay of amiR and siRRSV transgenic plants challenged with RRSV viruliferous BPH.

Genotypes	Number of Transplanted Plants	Number of Dead Plants	Number of Observed Plants	Number of Symptomatic Plants	Proportion of Symptomatic Plants
TN1	110	1	109	49	44.95%
Nip	106	3	103	38	36.89%
pC:empty	110	11	99	39	39.39%
pC:amiR1_2.1_	110	10	100	38	38%
pC:amiR1_20.1_	110	14	96	39	40.63%
pC:amiR2_2.2_	16	0	16	1	6.25% *
pC:amiR2_22.4_	110	5	105	21	20% *
pC:amiR1/2_9.1_	110	10	100	33	33%
pC:amiR1/2_12.3_	108	2	106	35	33.02%
pANDA:siRRSV_39.9_	104	0	104	36	34.62%
pANDA:siRRSV_40.3_	105	0	105	7	6.67% *
pANDA:siRRSV_41.9_	103	4	99	2	2.02% *

Asterisks indicate significant differences (*p* < 0.05 by Chi-square test) compared to the pC:empty and Nip controls for the amiR and siRNA constructs, respectively.

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
