# Peer review of "Optimized RNA-Silencing Strategies for Rice Ragged Stunt Virus Resistance in Rice"

_plants, 2021, doi:10.3390/plants10102008_

Round 1
Reviewer 1 Report
The paper is well writed, the introduction is complete and nice to read, the results well presented and discussed, with only some comments that could help the interpretation of the result (statistical analisys of disease incidence) and about line pc:amiR22.2

Author Response
Please see the attachements.

Reviewer 2 Report
Review of “Optimizing RNA …” by Lacombs et al.
General comment
The manuscript by Lacombs et al. describes production of transgenic rice plants which can express amiRNA and siRNA that potentially contribute to resistance to RRSV. The results showed that two of the transgenic lines expressing siRNA against RRSV appeared to be resistant to RRSV. The results support the conclusion by the authors, but the manuscript can be improved and the conclusion can be more concrete with additional data (see below). The quality of language is less than acceptable – too many grammatical errors, poor choice of words, awkward, confusing and illogical sentences, etc. Some of them are pointed out below (there are much more than this reviewer pointed out). The manuscript needs to be READ WELL, checked and re-worded thoroughly and entirely by the authors before it is edited by a professional.
Suggestions for improvement – this reviewer is not asking for additional data for the following, but the authors should touch upon the following issues in the manuscript.
- The source of RRSV should be examined thoroughly – The authors confirmed the infection of RRSV in the virus source plants collected from fields by RT-PCR. However, mixed infection of multiple viruses in rice in fields are not uncommon. Thus, the RRSV source plants should have been examined for the presence of other viruses found in the Mekong Delta.
- The height reduction rate in individual plants – The phenotyping for resistance to RRSV was done only by computing the proportion of symptomatic plants among those inoculated with RRSV. Measuring the reduction in the height in the transgenic plants, as it was done for non-transformed TN1 and Nipponbare, would be more informative and helpful to characterize the resistance phenotypes of transgenic plants.
- The presence of RRSV in inoculated plants – Examination for the presence of RRSV in inoculated plants is desirable to confirm the resistance or susceptible phenotypes of transgenic plants.
- Control plants for RRSV phenotyping – It was not clearly mentioned that what is “non-infected healthy plant” during the experiments. The controls should be plants which are inoculated with virus-free BPH since the infestation by BPH per se may affect the development of plants.
Specific comments – there are much more to fix than this reviewer pointed out below.
Line 23: could rapidly evolved >> could rapidly evolve or could have rapidly evolved
Line 38: will >> may
Line 56 and other lines: encodes for >> encodes (encode is a transitive verb)
Line 59: RRSV genome >> the RRSV genome
Line 62: a RNA >> an RNA
Line 102: were >> was
Line 104: chimeric >> a chimeric
Line 113: introduce >> introduced
Line 133: infection >> inoculation
Line 137: 2.1 Validation of >> what was validated in this section. This section describes how the amiRNA was designed.
Line 166: reported without impairing >> ?? reported not to have impaired??
Line 190: exclude that >> exclude the possibility that
Lines 238 to 270: Describe briefly with what the transgenic plants were transformed. This section is hard to understand because the description for the individual transgenic plants was not given before or in this section (the details for the transgenic plants was given in “Materials and methods” that appears in the last part of the manuscript”).
Line 291: Describe more clearly “non-infected condition”
Line 362: was shown >> had been previously shown
Line 444: constructions >> constructs
Line 446: where >> were
and so on….
Author Response
1- The source of RRSV should be examined thoroughly – The authors confirmed the infection of RRSV in the virus source plants collected from fields by RT-PCR. However, mixed infection of multiple viruses in rice in fields are not uncommon. Thus, the RRSV source plants should have been examined for the presence of other viruses found in the Mekong Delta.
We agreed this comment. We verified the absence of RGSV (Rice grassy stunt virus) as this virus is also found in Mekong Delta and transmitted by BPH. We added this information in the material and methods section of the revised version. We did not verify for the absence of other viruses found in Mekong Delta such as RTYV (Rice transitory yellowing virus) or RTV (Rice tungro virus) because these viruses are not transmitted by BPH.
2- The height reduction rate in individual plants – The phenotyping for resistance to RRSV was done only by computing the proportion of symptomatic plants among those inoculated with RRSV. Measuring the reduction in the height in the transgenic plants, as it was done for non-transformed TN1 and Nipponbare, would be more informative and helpful to characterize the resistance phenotypes of transgenic plants.
We completely agreed this comment. However, due to the important number of observed plants (more than 1000 plants), the individual measuring was not possible. As the important size reduction observed in nippombare infected plants, we assumed that visual observation would be satisfying.
3- The presence of RRSV in inoculated plants – Examination for the presence of RRSV in inoculated plants is desirable to confirm the resistance or susceptible phenotypes of transgenic plants.
We completely agreed this comment. Unfortunately as mentioned above, the important number of observed plants was not compatible with molecular characterization such as RT-PCR.
4- Control plants for RRSV phenotyping – It was not clearly mentioned that what is “non-infected healthy plant” during the experiments. The controls should be plants which are inoculated with virus-free BPH since the infestation by BPH per se may affect the development of plants.
We agreed that virus-free BPH inoculation would be a more accurate control. However, due to the important size reduction in the infected plant (more than half), we assumed that it could not be due only to the BPH infestation but to the virus infection.
Specific comments
Corrections directly in the attached revised version
The manuscript needs to be READ WELL, checked and re-worded thoroughly and entirely by the authors before it is edited by a professional.
The manuscript was already edited by a professional (American journal experts: https://www.aje.com). The three out of the four other reviewers only proposed moderate changes. The last reviewer do not feel qualified to judge

Reviewer 3 Report
It was a pleasure for me reading this manuscript talking about RNA silencing-based strategies
To my opinion, it would be better improving some parts of the manuscript.
The introduction is excessively long and at some points the countless details make the reader confused, one could think this is a review. Details are not always required to deal with the following experiments.
The authors should standardize the font size in the Figures, not always the same size.
2.1 Validation of selected amiR.
It not totally clear if the authors identified only 2 amiR (as described 1 and 2) and it is not totally clear if the authors analysed the sequences identity by sequence alignment before or after the alignment to amR1 and amiR2.
2.2 Validation of the functionality…
The validation of the functionality of miRNA precursor is truly interesting and clever but it remains an end in itself because not related to the real experiment with transgenics. That could be illustrated in a better way.
Line 228: it is difficult toundersatnd what that 20 out of …. stands for , because in the sequence HM125567 11 different mismatches are present and in the sequence L38900 other 11 mismatches as well.
Figure 4: it would be more appreciable to see different base colors. The resolution of the picture could be improved.
2.4 Accumulation of expected amiR….
It is not straightforward understand the full experiment: to this aim it would be better describe how many plants were analysed to study the accumulation of amiR and siRNA.
Lines 305 to 310: could the transgene have effect on the low seed germination or number of seeds?
The authors can explain that aspect.
2.5 Evaluation of RRSV resistance
The evaluation of the resistance to the virus is very interesting but the autors should clarify the meaning of the measurements done on the height of the plants
Author Response
The introduction is excessively long and at some points the countless details make the reader confused, one could think this is a review. Details are not always required to deal with the following experiments.
As this point was not mentioned by the other four reviewers, we did not change by introduction in the revised version.
The authors should standardize the font size in the Figures, not always the same size.
This was done in the attached revised figures.
2.1 Validation of selected amiR.
Please, find the changes in this result part of attached revised version.
2.2 Validation of the functionality…
Please, find the changes in this result part of attached revised version.
Figure 4: it would be more appreciable to see different base colors. The resolution of the picture could be improved.
The figure 4 is colored in the original version.
2.4 Accumulation of expected amiR….
Please, find the changes in this result part and in"evaluation of RRSV resistance"
of attached revised version.
2.5 Evaluation of RRSV resistance
Clarification is done concerning phenotypic evalutation in this part of the revised version.

Reviewer 4 Report
The authors reported two RNA silencing approaches (transgenic plants producing artificial miRNA or dsRNA) to induce resistance to Rice ragged stunt virus (RRSV) in rice. The techniques adopted and described are not innovative, the effects of amiRNA and siRNAs on viruses have been known for more than ten years, but for RRSV and rice it seems to be an approach that has not been previously studied.
The work has serious limitations in the number and choice of transgenic plants analysed, which unfortunately do not make the results completely reliable.
When using transformed plants it is advisable to analyse at least 3, better 5, independent transformation events to try to overcome a number of problems that the authors of this article have also had to face. Furthermore, the events chosen must all be functional, that is, if for example, my construct has the ultimate goal of overexpressing a gene, all the chosen lines must be overexpressing this gene, the silenced lines must be discarded. The authors used only 2 independent transgenic lines for amiRNA constructs, and in both lines amiR1 no amiRNA was detected. The explanations reported by the authors (lines 249-256) are not convincing because amiR1 is an active construct as demonstrated in N. benthamiana in Fig 3, and transgenic localization in the heterochromatin region is unlikely, since the gene for selection in rice it worked. The line pC:amiR2/2.2 with problem in germination (line 305) should be discarded. The authors should select other transformation events. Sometimes with miRNA overexpressing constructs, two copies of transgene are more functional than plants in single copy.
For the construct producing siRNAs, only 1 line worked well, the other two have to be discarded as one does not work at all and in the second line only with the S6 probe it is possible to see a very weak signal. Please use another probe for S6 fragment, the signal is too weak. Also in this case the authors' explanations are not very convincing: if the 3 fragments are cloned in tandem and therefore with inverted repeated orientation (line 398) the dsRNA would be formed for all fragments or for none (line 264). See above for consideration of heterochromatic localization (line 269). The authors should select other transformation events.
The authors used only transgenic plant with pC empty vector as control (line 303), but they used 2 different vectors for amiRNA and dsRNA constructs.
Authors should also evaluate symptom intensity not just presence / absence (Table 1).
Please add the statistical analysis in Table 1. How many times has viruliferous BPH inoculation been replicated? Only one?
Lines 331-356 and the discussion should be revised after characterization of other transgenic events and statistical analysis.
Line 75: only 21nt ?
Author Response
When using transformed plants it is advisable to analyse at least 3, better 5, independent transformation events to try to overcome a number of problems that the authors of this article have also had to face.
We agreed this comment. However, in our case phenotypic test to evaluate for RRSV resistance is quite a heavy procedure. Indeed RRSV infection has to be done through BHP vector transmission. According to expected transmission rate around 40%, we had to analyze around 100 plants for each events. That is why we could not select more than two independent plants for each event.
The authors used only 2 independent transgenic lines for amiRNA constructs, and in both lines amiR1 no amiRNA was detected. The explanations reported by the authors (lines 249-256) are not convincing because amiR1 is an active construct as demonstrated in N. benthamiana in Fig 3, and transgenic localization in the heterochromatin region is unlikely, since the gene for selection in rice it worked.
We agreed this comment. However, as mentionned in the manuscript, our first hypothesis to explain amir1 non-functionality is "The amiR1 construct could have been partially destroyed during rice transformation" (line 254). We agreed to remove the heterochromatin hypothesis (please see the attached revised version).
The line pC:amiR2/2.2 with problem in germination (line 305) should be discarded.
We kept the pC:amir2/2.2 in our analyses to illustrate that not only strong RRSV resistance is expected but also good agronomic abilities such as good germination rate and normal plant development.
For the construct producing siRNAs, only 1 line worked well, the other two have to be discarded as one does not work at all and in the second line only with the S6 probe it is possible to see a very weak signal.
We kept all these lines to illustrate the correlation between RRSV resistance and siRNA accumulation. The pANDA:siRRSV/39.9 line that do not accumulate any siRNA is a strong negative control. The pANDA:siRRSV/40.3 line that accumulate only S6 siRNA demonstrate that this single accumulation is enough to induce RRSV resistance.
Please use another probe for S6 fragment, the signal is too weak.
The probe used corresponds to the entire150 nt S6 segment. We cannot design other probe. Weak siRNA signals are reported in literature for similar transgenic plant samples ([22,32] for example).
Also in this case the authors' explanations are not very convincing: if the 3 fragments are cloned in tandem and therefore with inverted repeated orientation (line 398) the dsRNA would be formed for all fragments or for none (line 264). See above for consideration of heterochromatic localization (line 269).
As reported in the text, our first hypothesis is "a partial degradation of the transgenic construct during rice transformation" (line 268). We agreed to remove the heterochromatin hypothesis (please see the attached revised version).
The authors used only transgenic plant with pC empty vector as control (line 303), but they used 2 different vectors for amiRNA and dsRNA constructs.
The pANDA empty vector carries the ccdB gene encoding toxic protein. First, we used non-transformed Nipponbarre line as control. Then, the pANDA:siRRSV/39.9 line that do not accumulate any siRNA could be considerate as negative control.
Authors should also evaluate symptom intensity not just presence / absence (Table 1).
We completely agreed this comment. However, due to the important number of observed plants (more than 1000 plants), the individual measuring was not possible. As the important size reduction observed in nippombare infected plants, we assumed that visual observation would be satisfying.
Please add the statistical analysis in Table 1.
Statistical analysis has been added in the attached revised version.
How many times has viruliferous BPH inoculation been replicated? Only one?
Because of the quite heavy RRSV infection procedure and the important number of tested plants (more than 1000), only one replication was performed.
Line 75: only 21nt ?
The sentence has been changed to "miRNAs are mainly 21 nucleotides (nt) RNA molecule produced from an endogenous RNA precursor displaying a double strand hairpin structure containing miRNA and its inverse complementary sequence, miRNA*."

Reviewer 5 Report
Plant viral diseases cause important agricultural losses and are major threats to food security; rice ragged stunt virus (RRSV) is a major pathogen of rice. Lacombe et al. describe an RNA silencing strategy to generate transgenic rice lines with enhanced RRSV resistance. Artificial miRNA and inverted repeat constructs were compared in stable transgenic lines; a line expressing the inverted repeat construct showed robust resistance to RRSV. Of note the authors validated the use of the rice MIR159 gene as a precursor for the expression of multiple artificial miRNAs.
Although the use of RNA silencing is an established technique, this is a well-written manuscript that describes potential important resources for rice farmers, and it would thus be an interesting contribution to the journal.
I nonetheless suggest to revise some aspects to enhance the manuscript:
- To strengthen the findings, it could be helpful to accompany Table 1 with a plot that shows plant height of key lines and controls in mock and RRSV conditions, e.g. pANDA:siRRSV41.9 versus Nipponbare.
- Fig. 6, please clarify whether the symbols represent mean or median values, and error bars represent SD or SEM.
- Antiviral strategies through co-expression of multiple artificial RNAs have been reviewed and it would be useful to discuss them [1], including the use of artificial trans-acting siRNAs for expression of up to six artificial miRNAs from a single construct.
- Recent studies have revealed an emerging crosstalk between plant hormone signaling and RNA silencing pathways, which are altered during viral infection [2–4]. In this regard and in contrast to RNA silencing, strategies based on CRISPR/Cas systems do not rely on plant endogenous components and they have been reported for target viral RNA targeting also in rice, e.g. [5] and reviewed by [6]. These aspects could be discussed.
- Please revise, l. 13 ‘deployed to fight’; l. 54 ‘In GenBank’; l. 93, ‘double strand DNA constructs’ is unclear since DNA is naturally found in the double-stranded form, it might be replaced in the manuscript with ‘inverted repeat DNA constructs’; l. 121 not sure if ‘(Lafforgue et al. 2013)’ should be replace by a citation number; l. 223 ‘randomly chosen.’; l. 411 ‘resistance, here, the’.
1. Cisneros et al. Plants. 2020;9: E669. doi:10.3390/plants9060669
2. Zhang et al. Mol Plant. 2016;9: 1302–1314. doi:10.1016/j.molp.2016.06.014
3. Pasin et al. Plant Commun. 2020;1: 100099. doi:10.1016/j.xplc.2020.100099
4. Yang et al. Cell Host Microbe. 2020. doi:10.1016/j.chom.2020.05.001
5. Zhang et al. Plant Biotechnol J. 2019;17: 1185–1187. doi:10.1111/pbi.13095
6. Zhao et al. Plant Biotechnol J. 2020;18: 328–336. doi:10.1111/pbi.13278
Author Response
To strengthen the findings, it could be helpful to accompany Table 1 with a plot that shows plant height of key lines and controls in mock and RRSV conditions, e.g. pANDA:siRRSV41.9 versus Nipponbare.
We agreed reviewer comment that plant height measurement would illustrate pANDA:siRRSV41.9 resistance. However, because the size reduction due to RRSV infection is clearly, we assumed that visual observation would be satisfying enough.
Fig. 6, please clarify whether the symbols represent mean or median values, and error bars represent SD or SEM.
We added the following sentence in Figure 6 legend: "Means of size plants are reported for each conditions and error bars represent standard deviation."
Antiviral strategies through co-expression of multiple artificial RNAs have been reviewed and it would be useful to discuss them [1], including the use of artificial trans-acting siRNAs for expression of up to six artificial miRNAs from a single construct.
We added the reference in the discussion "As reviewed recently in Cisneros and Carbonell [52], strategies based on miRNA, Trans-acting siRNA and siRNA are strongly expected to produce simultaneously several artificial small RNA. Double miR and multiple siRNA techniques set up here can be considered as ones of these improved strategies and could be exploited in a more general context than virus resistance to simply and efficiently silence multiple target genes in rice." The following references numbers have been changed in the attached revised version
Recent studies have revealed an emerging crosstalk between plant hormone signaling and RNA silencing pathways, which are altered during viral infection [2–4]. In this regard and in contrast to RNA silencing, strategies based on CRISPR/Cas systems do not rely on plant endogenous components and they have been reported for target viral RNA targeting also in rice, e.g. [5] and reviewed by [6]. These aspects could be discussed.
Our mains results here are 1- the development of RRSV resistant rice line and 2- the setup of new techniques for multiple artificial small RNA expression in rice. We completely agreed reviewer comments about the great potential of CRISPR/Cas systems as antiviral strategy. However, we did not want to talk about this important topic not to dilute the main results of our work.
Please revise, l. 13 ‘deployed to fight’; l. 54 ‘In GenBank’; l. 93, ‘double strand DNA constructs’ is unclear since DNA is naturally found in the double-stranded form, it might be replaced in the manuscript with ‘inverted repeat DNA constructs’; l. 121 not sure if ‘(Lafforgue et al. 2013)’ should be replace by a citation number; l. 223 ‘randomly chosen.’; l. 411 ‘resistance, here, the’.
All these changes have been done in the attached revised version.

Round 2
Reviewer 4 Report
I understand the authors' answers well. But unfortunately they did not carry out any of the new analyzes requested also because the technical times of the reviews on the MDPI journals would not have allowed it. Unfortunately I have to confirm my previously expressed requests. New analyzes and a new submission in 6 months or a year would make the job excellent. I am also aware that other reviewers would accept the work as it is now, but I would advise authors not to publish such an article, when with more work they could publish a truly outstanding article.
Author Response
New analyzes and a new submission in 6 months or a year would make the job excellent.
We agree with reviewer comment. However, as accepted by the four other reviewers, we do not plan to perform additional experiments. We wish to publish our results such as this new version with improved English corrections.